# Spatial distribution of poultry farms using point pattern modelling: A method to address livestock environmental impacts and disease transmission risks

**Marie-Cécile Dupas**[1,2], **Francesco Pinotti**[3], **Chaitanya Joshi**[4], **Madhvi Joshi**[4], **Weerapong Thanapongtharm**[5], **Madhur Dhingra**[6], **Damer Blake**[7], **Fiona Tomley**[7], **Marius Gilbert**[1], **Guillaume Fournié**[7,8,9]*

**1** Spatial Epidemiology Lab, Université Libre de Bruxelles, Brussels, Belgium, **2** Data Science Institute, Hasselt University, Hasselt, Belgium, **3** University of Oxford, Oxford, United Kingdom, **4** Gujarat Biotechnology Research Centre, Gandhinagar, India, **5** Department of Livestock Development, Ministry of Agriculture and Cooperatives, Bangkok, Thailand, **6** Emergency Prevention system for Animal Health, Food and Agriculture Organization of the United Nations, Rome, Italy, **7** Department of Pathobiology and Population Sciences, Royal Veterinary College, London, United Kingdom, **8** INRAE, VetAgro Sup, UMR EPIA, Université de Lyon, Marcy l'Etoile, France, **9** INRAE, VetAgro Sup, UMR EPIA, Université Clermont Auvergne, Saint Genes Champanelle, France

* gfournie@rvc.ac.uk

**Data Availability Statement:** The codes and data are available on github: https://github.com/mcdupas/FDM. The poultry farm distributions used

## Abstract

The distribution of farm locations and sizes is paramount to characterize patterns of disease spread. With some regions undergoing rapid intensification of livestock production, resulting in increased clustering of farms in peri-urban areas, measuring changes in the spatial distribution of farms is crucial to design effective interventions. However, those data are not available in many countries, their generation being resource-intensive. Here, we develop a farm distribution model (FDM), which allows the prediction of locations and sizes of poultry farms in countries with scarce data. The model combines (i) a Log-Gaussian Cox process model to simulate the farm distribution as a spatial Poisson point process, and (ii) a random forest model to simulate farm sizes (i.e. the number of animals per farm). Spatial predictors were used to calibrate the FDM on intensive broiler and layer farm distributions in Bangladesh, Gujarat (Indian state) and Thailand. The FDM yielded realistic farm distributions in terms of spatial clustering, farm locations and sizes, while providing insights on the factors influencing these distributions. Finally, we illustrate the relevance of modelling realistic farm distributions in the context of epidemic spread by simulating pathogen transmission on an array of spatial distributions of farms. We found that farm distributions generated from the FDM yielded spreading patterns consistent with simulations using observed data, while random point patterns underestimated the probability of large outbreaks. Indeed, spatial clustering increases vulnerability to epidemics, highlighting the need to account for it in epidemiological modelling studies. As the FDM maintains a realistic distribution of farm location and sizes, its use to inform mathematical models of disease transmission is particularly relevant for regions where these data are not available.

in the document are published for Gujarat (State of India). Data for Thailand can be requested by contacting saraban@dld.go.th, and data for Bangladesh can be requested by contacting FAO-BD@fao.org.

**Funding:** o F.T. is supported by the UKRI GCRF One Health Poultry Hub (Grant No. B/S011269/1), one of twelve interdisciplinary research hubs funded under the UK government's Global Challenge Research Fund Interdisciplinary Research Hub initiative. G.F. is supported by the French National Research Agency and the French Ministry of Higher Education and Research. The funders had no role in study design, data collection and analysis, decision to publish, or preparation of the manuscript.

**Competing interests:** The authors have declared that no competing interests exist.

### Author summary

We have developed a model to predict the location and size of poultry farms in countries or regions with limited data. This is important because knowing the distribution of farms helps in understanding how diseases spread, especially in areas with rapidly growing farm populations. Our model uses advanced statistical methods and is calibrated with environmental and human activity data to simulate farm locations and sizes, which we tested on farms in Bangladesh, Gujarat (India), and Thailand. We found that our model creates realistic patterns of farm locations and sizes, which are crucial for predicting disease outbreaks. When we simulated disease spread, our model showed that farms clustered together are more vulnerable to large outbreaks. This highlights the need for realistic farm data in disease prevention efforts. Our model can help public health officials in regions without detailed farm information to better plan and respond to potential disease threats. This work is a step towards better protecting both animal and human health from the spread of diseases.

## 1 Introduction

Livestock contribute to food security as a main source of animal protein (dairy, meat and eggs) and provide 17% of the world population's dietary energy intake [1] whilst occupying 80% of global agricultural land [2]. The consumption of animal source food is increasing most rapidly in low- and middle-income countries (LMICs) [3]. In Asia, chicken meat production has quadrupled in the last two decades [4]. This rapid intensification has been characterised by a shift in the spatial distribution of farms, particularly pig and poultry farms which have smaller land requirements than ruminants, transitioning from rural to peri-urban areas [5–7].

Accurate and up-to-date maps of livestock farms are crucial to assess their environmental impacts and the risk of diseases spreading through livestock populations [8]. However, recording exact farm locations may not be achievable in many LMICs due to limited available resources and may also be constrained for privacy reasons, including in HICs. For this reason, some studies have relied on modelling frameworks to estimate high-resolution farm distribution at various administrative levels, and to improve on coarse census data. Such frameworks have mostly employed linear regression models [9–11]. Van Boeckel et al. [12] mapped the distribution of intensive poultry farms in Thailand using a simultaneous autoregression model (SAR) that explicitly accounted for spatial autocorrelation. However, as reported by the authors, the model failed to capture the high levels of spatial clustering that are observed among intensive farms in that country. Random forest models have been shown to outperform linear regression models when used to downscale census data at the global scale for several livestock animal species [13] or at national scale for pig populations in Thailand [14] and China [15]. The use of remote sensing combined with synthetic data methods results in realistic distribution and density patterns of farms in the United States [16]. However, the distribution of animals at farm level and the process of generating clustered point distributions are yet to be embedded in these models. At the global level, the Gridded Livestock of the World (GLW) predicts livestock as a continuous, gradually varying, density of animals per pixel at 10 km or 1 km resolution [10, 13, 17]. Thus, these models do not provide information about how animals are distributed across farms, and how farms are distributed across space, despite these parameters having major influence on both environmental impacts and disease risk associated with intensification of livestock production.

This paper develops a novel modelling framework that alleviates limitations of previous models and can be used to predict both farm locations and sizes. The framework builds on a previous point pattern model introduced by [18, 19] that was used to predict clustered farm distributions. We show that this Farm Distribution Model (FDM) successfully predicts spatial farm locations and sizes of poultry farms in three 'test' geographic regions (Bangladesh, Gujarat and Thailand) that are characterised by different levels of intensification and for which farm data were already available.

We trained a Log Gaussian Cox Process (LGCP) and Random Forest model (RF) and assessed their external and internal validity using three observed point patterns. This allowed us to test the robustness of the method to reproduce farm distribution in data-scarce countries. We further illustrate the relevance of our approach to inform models of disease spread in livestock by comparing epidemic simulations on empirical and synthetic farm distributions generated with different methods.

## 2 Methods

### 2.1 Training data sets

The modelling procedure was based on farm size and location data from three regions in Asia: Thailand and Bangladesh (whole country) and Gujarat (state in India). The Gujarat Biotechnology Research Centre collected data on the distribution of farms in Gujarat, India. The data represented are based on the information acquired from the Department of Animal Husbandry, Dairying & Fisheries, Ministry of Agriculture & Framers Welfare, Government of India and Directorate of Animal Husbandry, Government of Gujarat, Gandhinagar, Gujarat, India in the year 2020.

Around 59% of farm locations in Gujarat corresponded to village centroids coordinates. Farms with overlapping locations were assigned to random points within the area of the corresponding villages. A similar procedure was adopted for farms in Thailand for which only the village location is known (data collected in 2010 by the Department of Livestock Development [18]). Finally, the geographic coordinates of the farms in the Bangladesh dataset were obtained from an agricultural census collected by the Food and Agriculture Organization of the United Nations, ensuring accuracy and reliability of the location data.

Bangladesh and Gujarat data sets cover areas of similar size, while farms in Thailand are scattered over a region that is around three times larger (Table 1). Data sets consisted of the coordinates and capacity of farms differentiated according to their type of production into broiler and layer farms (S1 Fig). We assumed that the size of a farm coincides with its capacity (i.e. maximum number of animals that can be raised on a farm) and hence ignored yearly stock variations. We kept only farms with more than 500 chickens since the original FDM was developed for intensive farms [18].

**Table 1. Characteristics of data sets in terms of area surface, intensity of points distribution (points/m$^2$) and economic features.** GDP is given for 2019 for the three regions. The distribution of broiler and layer chicken farms in Gujarat, Thailand and Bangladesh was collected in 2020, 2010 and 2010 respectively.

| Area | Area Code | GDP[1] (billion US$) | Surface area ($10^3$ km$^2$) | Intensity (pts/m$^2$) | Intensity of broiler farms (pts/m$^2$) | Intensity of layer farms (pts/m$^2$) | Number of farms (broilers—layers) |
|---|---|---|---|---|---|---|---|
| Gujarat | IN.GJ | 230 | 196 | $1.77 \times 10^{-8}$ | $9.88 \times 10^{-9}$ | $1.39 \times 10^{-9}$ | 2,611—311 |
| Thailand | THA | 543.5 | 514 | $4.81 \times 10^{-9}$ | $3.14 \times 10^{-9}$ | $1.03 \times 10^{-9}$ | 3,717—1,439 |
| Bangladesh | BGD | 302.6 | 135 | $2.27 \times 10^{-7}$ | $9.80 \times 10^{-8}$ | $3.75 \times 10^{-8}$ | 22,159—9,074 |

[1] worlbank 2019

**Table 2. List of Spatial Predictors with their sources and units used in the LGCP and RF models.**

| Type | Variable | Units | Source | Abbreviation |
|---|---|---|---|---|
| Anthropogenic | Human population density | Log10 people per hectare | Tatem, 2017 [20] | Hpop |
| | Proximity to cities with 5,000,000<x<50,000,000 inhabitants | Minute$^{-1}$ | Nelson et al., 2019 [21] | Access_MC1 |
| | Proximity to cities with 50,000<x<50,000,000 inhabitants | Minute$^{-1}$ | Nelson et al., 2019 [21] | Access_MC11 |
| | Proximity to cities with with 1,000,000<x<5,000,000 | Minute$^{-1}$ | Nelson et al., 2019 [21] | Access_MC2 |
| | Proximity to large and medium ports | Minute$^{-1}$ | Nelson et al., 2019 [21] | Access_Port12 |
| | Proximity to roads | Minute$^{-1}$ | Meijer et al., 2018 [22] | Proxim_roads |
| Topography | Slope | | Amatulli et al., 2018 [23] | Slope |
| Vegetation | Crop cover | Pixel % covered by crops | Fritz et al., 2015 [24] | Crop |
| | Tree cover | Pixel % covered by forest | Hansen et al., 2013 [25] | Tree |
| Livestock | Chicken population density | Log10 animals per hectare | Gilbert et al., 2018 [13] | nChicken |

## 2.2 Spatial predictors

Table 2 lists the spatial predictors used for the LGCP and RF models accross 4 categories of covariates: anthropogenic, topographical, vegetation and livestock characteristics. The distribution of chicken density was derived from the most recent version of the Gridded Livestock of the World (GLW, [13]). Proximity predictors were the inverse of time travel to major cities, ports and roads ($x = \frac{1}{timetravel+1}$), so that the maximal values were associated to the closest locations. These predictors allowed us to assess if farm locations were affected by infrastructure density. Other predictors were used as originally published (references in the Table 2).

## 2.3 Point pattern modelling

The procedure for modelling farm locations is based on the point pattern analysis method described in [19]. We modelled spatial point patterns using LGCP associated with spatial predictors and the Palm maximum likelihood method of parameters optimisation [26]. This approach was found to outperform other types of point pattern models at reproducing clustered farm distributions [18, 19]. This method is suitable to deal with highly inhomogeneous point intensity and spatial autocorrelation. In the LGCP model, point distributions are generated in space stochastically according to a Poisson process with intensity $\lambda(u)$, which is modeled as the exponential of a Gaussian random field:

$$\lambda(u) = exp(Z(u)) . \tag{1}$$

Here, $Z(u)$ represents the sum of a deterministic linear predictor based on spatial covariates and a stochastic Gaussian process, $Y(u)$:

$$Z(u) = (\theta_0 + \theta_1 \text{pred}_1 + \theta_2 \text{pred}_2 + \ldots + \theta_n \text{pred}_n) + Y(u) . \tag{2}$$

The Gaussian process, $Y(u)$, introduces spatial correlation among the points and is characterized by a zero-mean and a specified covariance structure, ensuring that the model captures the spatial dependency inherent in the geographic distribution of farms.

The spatial distribution of farms was modelled by estimating the intensity function $\lambda(u)$ on a 128x128 grid, where each grid cell represents a specific spatial location $u$. This configuration was automatically set by `spatstat` (version '2.3.3', [27]) to effectively capture spatial variations within the study regions [28].

We applied a LGCP model to each pairing of study region (Bangladesh, Thailand, and Gujarat) with a poultry production type (broiler or layer), resulting in a total of six models.

The validity of these models was then evaluated both within their respective training regions (internal validation) and by application to regions where they were not originally trained (external validation).

The importance of each spatial predictor $pred_i$ was computed as the product of its maximum value across space and its estimated weight $\theta_i$.

## 2.4 Point pattern characterisation and model validation

**2.4.1 Spatial correlation analysis of points pattern.**   Ripley's K-function measures the clustering behaviour in a spatial point pattern (SPP), and is defined as the cumulative average number of data points found within a distance $r$ of a typical data point [28, 29]. The inhomogeneous K-function, $K_{inhom}(r)$, is a generalization of Ripley's K-function designed to analyze point patterns with varying intensity across space. The inhomogeneous K-function is defined as follows:

$$K_{inhom}(r) = \frac{|W|}{N(N-1)} \sum_{i=1}^{N} \sum_{\substack{j=1 \\ j \neq i}}^{n} \frac{1}{\widehat{\lambda_i}\widehat{\lambda_j}} \mathcal{I}\{d_{ij} \leq r\} e_{ij} \,, \tag{3}$$

where $N$ is the number of points, $|W|$ denotes total study area, $\widehat{\lambda_i}$ and $\widehat{\lambda_j}$ are respectively the estimated intensity functions at point $i$ and $j$, $\mathcal{I}\{d_{ij} \leq r\}$ equals 1 if the euclidian distance $d_{ij}$ between points $i,j$ is less than $r$ and is 0 otherwise, and $e_{ij}$ is an edge correction weight to avoid sampling biases. Given the location of the first point $x$ and the distance $d = \|x - x'\|$, the second point $x'$ must lie in the circle $b$ of radius $d$ and centred at $x$. However, the circle $b$ is generally only partly inside the study area $W$ for large $d$. Then, the Ripley's isotropic correction $e_{ij}$ uses the fraction of the length of the circle, $\ell$, that is within the study area (S2 Fig) and considers that the point pattern is isotropic (statistically invariant under rotation). We calculated the probability of the second point $x'$ being inside the window $W$ as:

$$p(x, d) = \frac{\ell(W \cap \delta b(x, d))}{2\pi d}, \tag{4}$$

Finally, the edge correction is:

$$e_{ij} = \frac{1}{p(x_i, d_{ij})}. \tag{5}$$

We used the Besag's transform of the in-homogeneous K-function ($K(r)$) by using the function *linhom* in the package *spastat* which is:

$$L_{inhom}(r) = \sqrt{\frac{K_{inhom}(r)}{\pi}}. \tag{6}$$

**2.4.2 Global rank envelope test for model validation.**   The global rank envelope test is a robust statistical method used to evaluate the goodness-of-fit of a model by comparing the observed data to a collection of simulated data generated from the model. It provides a comprehensive approach for assessing whether simulated and observed patterns are consistent [30].

The global rank envelope test is based on a chosen test statistic, in this study the $L_{inhom}(r)$ function, a transformed version of the inhomogeneous K-function. For each simulation and at each distance $r$, we computed the test statistic and ranked the observed value of the statistic

among the simulated values. This created an envelope of expected values under the model. If the observed test statistic lay within this envelope for all distances, it suggested that the model was an adequate fit to the observed data. The test was performed using the `envelope` and `Linhom` functions from the `spatstat` package in R [27, 28]. The test procedure is described in the Appendix S1 File.

**2.4.3 Quadrat counting tests.**   We divided study areas into quadrats, and computed counts of points within each quadrat (n = 23 − 44 depending on the study area) [31]. The patterns of quadrats are shown in S3 Fig. We did not consider quadrats that occupy less than 80% of the complete theoretical polygon to avoid edge effects. The performance of a model was evaluated by computing the correlation coefficient between the log-transformed of the number of points per quadrat between the observed and simulated pattern patterns.

## 2.5 Farm size modelling: Random Forest model with spatial predictors

The second-step of the algorithm consists of training a RF regression model to predict farm sizes. First, we averaged spatial predictors within a radius of 5,000 m around each farm. We tested different buffer zone sizes, of 2,500 m, 5,000 m and 7,500 m, and we selected the 5,000 m buffer zone in the final analysis as it performed slightly better than others. Secondly, we transformed farm sizes $X$ using a power function to reduce the skewness of their distribution:

$$X_{transform} = \frac{X^a - 1}{a} \, , \tag{7}$$

and used the function *PowerTransformer* from the *sklearn* package in Python to fit the parameter $a$. We used the function *RandomForestRegressor* of the *sklearn* package in Python, with 500 decision trees.

The goodness of fit (GOF) metrics of the predictions were all established through cross-validation, i.e. by measuring the correlation between observed and predicted animal numbers in farms that were not used to train the Random Forest models. The total data set was divided into a training data set (75% of the data) and a validating data set (25%). This process was repeated 5 times, each time selecting a different random set of farms to train the RF models. We then calculated GOF measures, i.e. the correlation coefficient and the root-mean-square error (RMSE) between predicted and observed farm sizes for each fold. Both GOF measures are calculated using log transformed and absolute values of farm sizes.

## 2.6 Mathematical modelling of disease transmission

**2.6.1 Simulations.**   We simulated the spread of a pathogen over $M$ poultry farms with spatial coordinates $(x_i, y_i)$ and sizes $X_i$, $i = 1, \ldots, M$. Simulations were stochastic and farms' infection statuses were updated synchronously, with each time step being 1 day long. Each farm was either susceptible to infection ($S$), infectious ($I$) or removed ($R$). An infectious farm $i$ transmits the pathogen to a susceptible farm $j$ with daily probability:

$$p_{ij}(S \rightarrow I) = 1 - exp(-\gamma_{ij}) \, , \tag{8}$$

where the force of infection exerted by $i$ on $j$ is given by:

$$\gamma_{ij} = \beta \cdot X_i^{Q_I} \cdot X_j^{Q_S} \cdot K(d_{ij}) \, , \tag{9}$$

$\beta$ denoting transmissibility and $K(d_{ij})$ representing a spatial transmission kernel depending solely on the (euclidean) distance between $i$ and $j$. The exponents $Q_I$ and $Q_S$ allow for different scalings of the force of infection with the sizes of infectious and susceptible farms, respectively.

Infectious farms recover with daily probability:

$$p(I \rightarrow R) = 1 - exp(-\mu), \tag{10}$$

where $\mu$ is the recovery rate.

At the end of the infectious period, the removed state ($R$) describes premises that were culled following reporting of disease and/or anomalous excess deaths caused by the viral incursion (which we are not modelling explicitly). We assume that these farms are depopulated and remain empty until the end of the outbreak. Therefore, removed farms are not infectious and can not be reinfected in simulations. These assumptions are more appropriate to describe the spread of a highly pathogenic avian influenza virus strain such as H5N1.

We implemented our simulations in C++ using the *Conditional subsample* algorithm [32]. Briefly, the algorithm overlays a grid over the study area, so that transmission attempts involving farms belonging to different grid cells can be checked only after resolving whether any transmission occurs between those cells. In order to ensure an efficient implementation, we used a heuristic, adaptive routine to identify an optimal gridding. Both simulation and cell-construction routines are detailed in [32].

**2.6.2 Transmission kernels.** We considered a power-law transmission kernel employed by Hill et al. (2017) [33] to study the spread of H5N1 avian influenza virus in Bangladesh:

$$K(d) = 1 \quad \text{if} \quad 0 \leq d < d_{min}, \tag{11}$$

and

$$K(d) = \left( d_{min} / d \right)^{\alpha} \quad \text{if} \quad d \geqslant d_{min}, \tag{12}$$

with $d_{min} = 0.1km$.

We considered long-ranged and short-ranged transmission kernels corresponding respectively to $\alpha = 0.643$ and to $\alpha=3$ (S4 Fig). The former value was retrieved from Hill et al. (2017) [33], while the latter is compatible with shorter transmission lengths across farms as observed in earlier works [34].

**2.6.3 Simulation scenarios.** For a given study area, we simulated pathogen transmission using six different spatial distributions of farms. First, we considered the empirical distribution of farms, which we used as a reference. Results from this scenario were then compared with SPP generated from the LGCP, and a random distribution of points pattern generated according to a Complete Spatial Randomness process. We also compared a homogeneous farm size scenario, where all farm sizes were set to the average farm size (Constant Size; CS), with a heterogeneous farm size scenario, where RF model was used to assign a size to each farm (Random Forest Size; RFS). All scenarios are summarized in Table 3 and displayed in S5 Fig.

We ran 2000 independent simulations for each scenario. In the case of simulated farm distributions, we generated 40 independent farm distributions and ran 50 disease spreading simulations for each model formulation. In each simulation, we initialised the infection by selecting and infecting a random farm at $t = 0$; a simulation stopped when no infectious farms remained.

**2.6.4 Spatial epidemic risk.** In order to compare spatial predictions of epidemic risk using different farm distributions, we implemented the following methodology. We defined the risk $V_i$ for farm $i$ as the proportion of 100 simulations in which an epidemic starting from this farm reaches at least 100 farms. For each distribution of farms, we first calculated the risk $V_i$ for each farm. In the case of BGD we considered only 4000 random farms as initial seeds due to long computation times. Then, in order to compare different farm distributions, we defined a common spatial grid covering the study area and averaged risk $V_i$ in each cell. This

**Table 3. Farm distributions considered in the disease transmission modelling.**

| Name | Description |
|---|---|
| Empirical | Uses observed data |
| Empirical (CS) | Uses empirical locations but farm size is set to the average farm size for all farms. |
| Random+CS | Farms are scattered uniformly at random over the study area; farm size is set to a constant value (Constant Size; CS), namely average farm size. |
| LGCP+CS | Farm locations are generated from a LGCP; farm sizes are set to a constant value (Constant Size; CS), namely average farm size. |
| Random+RFS | Farms are scattered uniformly at random over the study area; farm sizes are generated from a RF model (Random Forest Size; RFS). |
| LGCP+RFS | Farm locations are generated from a LGCP; farm sizes are generated from a Random Forest model (Random Forest Size; RFS). |

procedure yielded an average risk $V_a$ for each cell in the cell $a = 1, \ldots, N_{cel}$. Using the same spatial grid for all point models, we then performed a quantitative comparison between the different models on the basis of $V_a$. We used a rectangular grid of 40x40 km cells to the empirical distribution of farms while allowing for an additional margin of 20 km in each direction.

To assess the extent to which the maps of $V_a$ obtained using LGCP models trained on different sites match the estimate of $V_a^{empirical}$ obtained by using the empirical distribution, we calculated the Spearman's rank correlation coefficient between $V_a^{empirical}$ and the maps $V_a(j)$, where $j = 1, 2, \ldots, 40$ extends over all the realisations generated from the same model (we omit any pair of cells where at least one does not contain farms). We thus obtain a collection of 40 correlation coefficients for each point model.

## 3 Results

### 3.1 Characterisation of spatial homogeneity of farms distribution

Farm density is higher in Bangladesh than in Gujarat and Thailand (Table 1). In all three study areas, the density of broiler farms is higher than for layer farms. According to the L-function, all empirical SPPs are more clustered than a random SPP for distances under 100 km (Fig 1A). The maximal level of clustering occurs under 20 and 25 km for all SPPs, except for layer farms in Bangladesh where it occurs at around 9 km. For all three study areas, layer farms are more clustered than broiler farms for around $r < \frac{r_{max}}{2}$, with $r_{max}$ being the maximum radius for each area.

We trained a LGCP model for each area and production type (6 models). Among all predictors, proximity to roads had the highest influence on the locations of farms, except for layer farms in Gujarat for which the distribution is affected by chicken density and tree cover (Fig 1B). Human density is important for all models, while other accessibility predictors, crop and slope were the least important covariates.

### 3.2 Performance of the farms location model (LGCP)

LGCP model generates a different simulated SPP at each simulation (S6 Fig). We evaluated the goodness of fit of the farm location models by two procedures. First, we assessed if the simulated and observed SPPs display similar inhomogeneous patterns by calculating the L-function (section 2.4.1). Second, the quadrat count test allowed us to assess if clusters of farms in the observed and simulated SPPs were similarly located across a study area (section 2.4.2). As an L-function was computed for each simulated SPP, we plotted the envelope of all L-functions

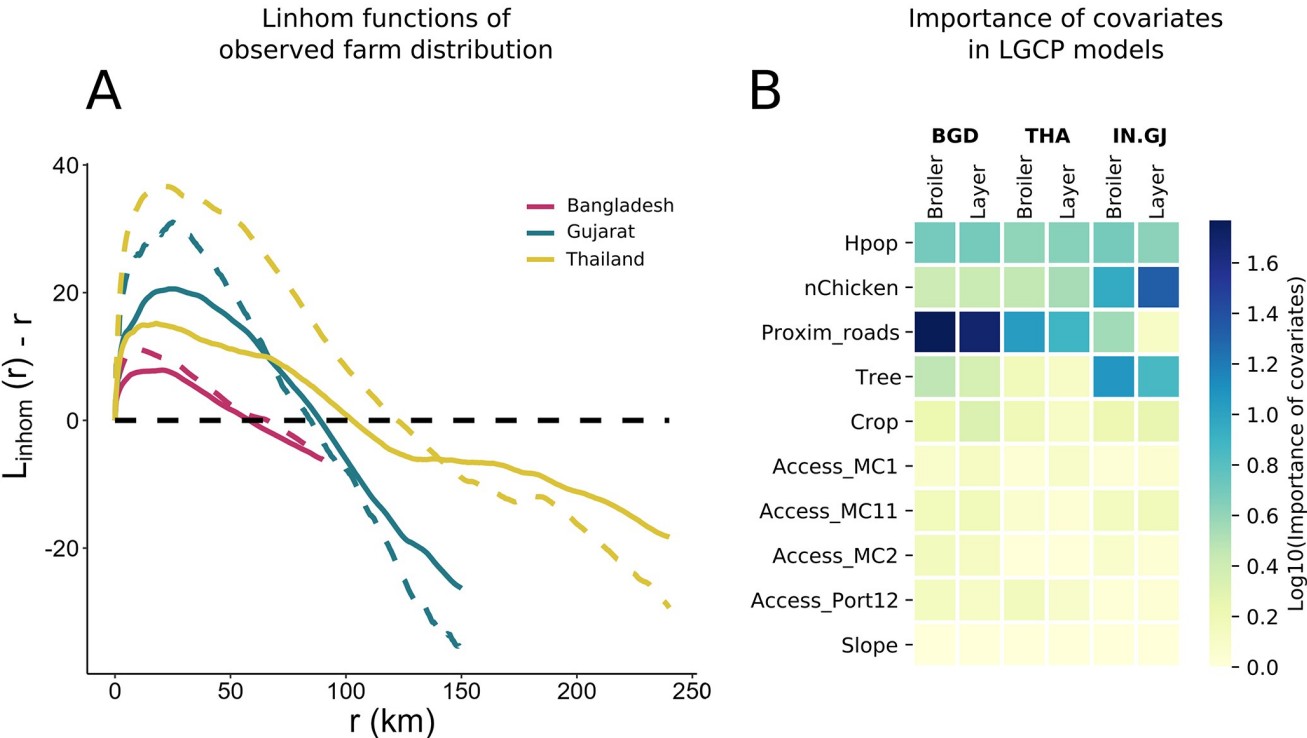

**Fig 1. A. L-function as a function of $r$ for each training data set**. The L-function for broiler (solid) and layer (dashed) farms are presented. Subtracting $r$ from $L_{inhom}(r)$ aids in interpreting the plots; when $L_{inhom}(r) - r$ equals zero, it signifies complete spatial randomness. Values of $L_{inhom}(r) - r$ greater than zero suggest a clustering pattern, whereas negative values indicate a dispersed or regular pattern relative to a random spatial point pattern at the scale of $r$. The black dashed line represents the L-function of a completely random point pattern. Points above this line denote more clustering, whereas points below indicate greater dispersion than would be expected under spatial randomness. **B. Importance of covariates for LGCP models on a logarithmic scale.** One model was trained per production type and study area (Bangladesh, Thailand, and Gujarat).

generated by 8000 simulations, and compared these to the L-function of the observed SPPs. Fig 2 shows the results for broiler farms.

**3.2.1 Broiler farms.** For broiler farms, the model trained using Bangladesh data offers the best prediction in terms of both internal and external validation. Indeed, the envelopes generated with the model trained on Bangladesh data and applied to Bangladesh and Gujarat include, or are near, the respective observed L-function (Fig 2). Although the Bangladesh model underestimates the clustering level of the Thailand SPP (Fig 2B), it reproduces the L-function for low radii of the Gujarat SPPs (Fig 2C) even though it is different from the Bangladesh L-function. Moreover, the Bangladesh-trained model locates the cluster better with high correlation coefficient between observed and simulated SPPs (Fig 3A).

Although the Gujarat model fails the global rank envelope test for internal validation (S7A and S7B Fig), the observed L-function remains close to the global envelope. We also note that the latter is particularly thin, indicating consistency between simulated SPPs. While the Gujarat model has a high p-value when applied to Bangladesh (S7A and S7B Fig), the global envelope is wide (Fig 2G), implying high variability in clustering between simulated SPPs. This suggests the need to interpret the p-value of the global envelope test in combination with the visualisation of the observed L-function and simulated envelope. In addition, the model locates clusters of farms in Thailand and Gujarat, but not in Bangladesh (Fig 3A). However, for simulated SPPs in Thailand, the global envelope test indicates a higher level of clustering for distances above 50 km than observed.

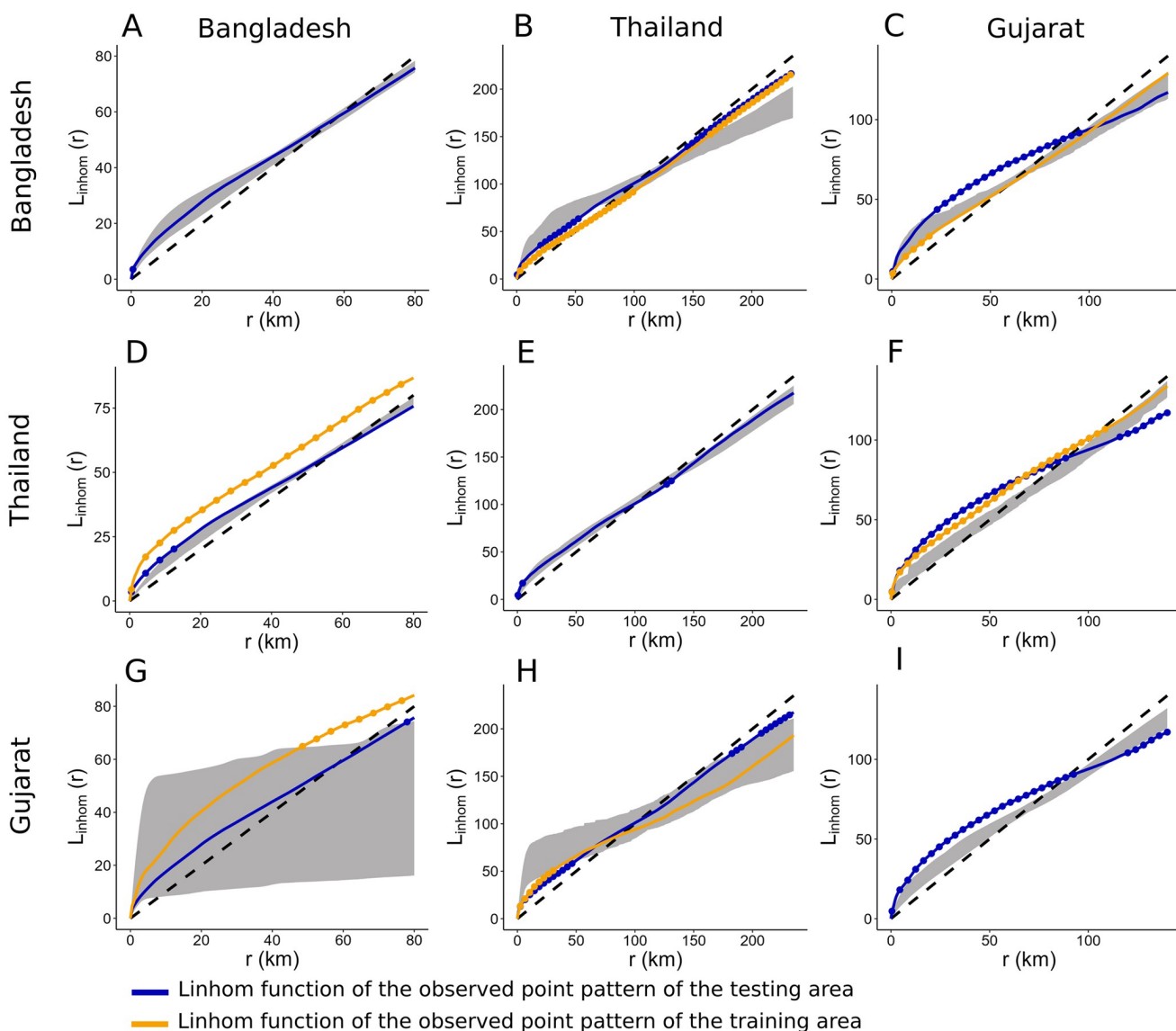

**Fig 2. Global envelope of the L-function of simulated points patterns with the LGCP model for broiler farms.** The envelope of the simulated SPPs is represented in grey. The L-function of the training point pattern is in orange, and the L-function of the observed points pattern of the testing area is in blue. Points outside the envelope are highlighted with dots. **A & B & C.** Envelope test of simulated SPPs generated with the model trained in Bangladesh, Applied in Bangladesh, Thailand, and Gujarat. **D & E & F.** Envelope test of simulated SPPs generated with the model trained in Thailand, Applied in Bangladesh, Thailand, and Gujarat. **G & H & I.** Envelope test of simulated SPPs generated with the model trained in Gujarat, Applied in Bangladesh, Thailand, and Gujarat.

Finally, the model trained in Thailand reproduces only its own spatial point patterns (Fig 2E), with a thin global envelope of simulations, indicating consistency between simulated SPPs.

**3.2.2 Layer farms.** All three models for layer farms satisfy the internal validation test of the global envelope (S7B and S7D Fig). Again, the model trained in Bangladesh performs best, with high global envelope p-value when applied to Bangladesh and Gujarat, and high quadrat correlation coefficient for the three areas. The model underestimates the level of clustering in Thailand, even though the envelope remains close to the observed L-function and followed the same trend.

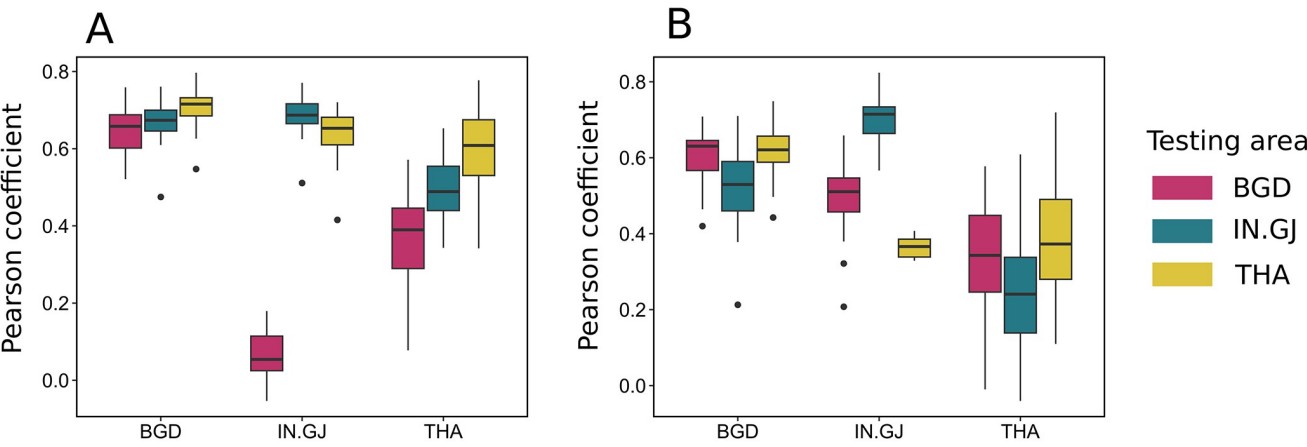

**Fig 3. Boxplot of the correlation coefficient between the numbers of points per quadrat in observed and each simulated SPP for each type of production: broiler (A) and layer (B).** Model names are indicated in the abscissa labels and refer to the area where the model was trained. The color of the boxes indicate where the model is tested (red for Bangladesh, blue for Gujarat and yellow for Thailand).

Although the Gujarat model is associated with high p-values when applied to Bangladesh and Gujarat, the envelopes are wide (Fig 4G). Also, the model does not reproduce the level of clustering and the locations of clusters in Thailand (Fig 3B).

Finally, the Thailand model reproduces the Bangladesh L-function, despite SPPs differing widely across countries. However, prediction of cluster locations is poor (Fig 3B).

### 3.3 Farm size predictions with random forest model

The second step of the FDM consists of predicting the farm sizes using a RF model conditioned by farm locations (generated during the first step). Two RF models were trained, one for broiler farms and another for layer farms. The training data set of these two models covered the three study areas. The most important predictors for farm size include proximity to major cities, tree cover, human and chicken population densities (S8 Fig). The distributions of the log size of farms are close to an unimodal distribution for all three data sets. Bangladesh, which has the largest number of farms of all three countries (Table 1), exhibits the lowest farm size peak with a median of 1,000 broiler chickens and 1,100 layer hens per farm. In contrast, Gujarat and Thailand, which are characterised by more intensive livestock production systems, have a median of respectively 5,000 and 10,000 chickens per broiler farm; and a median of 12,000 and 6,500 chickens per layer farm.

The RF model predicts log transformed size of farms with an average correlation coefficient of 0.83 and 0.70 over the five bootstraps for respectively broiler and layer farms (S1 and S2 Tables). The RMSE between log observed and predicted values is also weaker for broiler farms with around 0.275 against 0.343 for layer farms. These two GOF measures indicate a significant predictability of farm sizes through RF model. Moreover, the distribution of observed and predicted farm sizes shows that the RF model allows us to reproduce the high heterogeneity of the farm size range thanks to the log transformation (Fig 5). However, heterogeneity is not maintained when the RF model is applied to the distribution of farms generated with LGCP (S9 Fig).

### 3.4 Epidemic transmission modelling

We now present the results of disease transmission simulations. As detailed in the Methods section, we considered an array of 6 spatial farm distribution models with farm locations

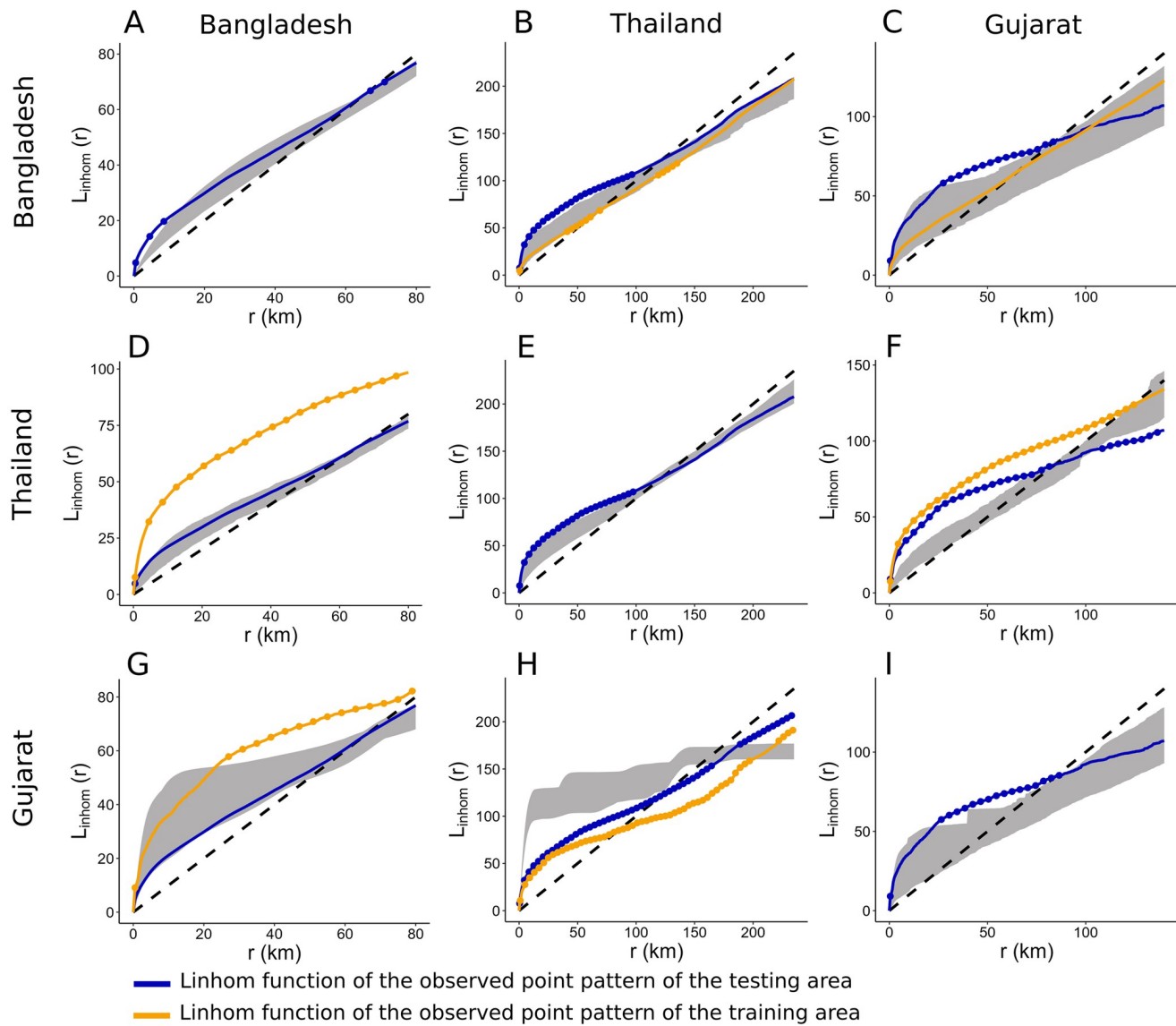

**Fig 4. Global envelope of the L-function of simulated points patterns for layer farms.** The envelope of the simulated SPPs is represented in grey. The L-function of the training point pattern is in orange, and the L-function of the observed points pattern of the testing area is in blue. Points outside the envelope are highlighted with dots. **A & B & C.** Envelope test of simulated SPPs generated with the model trained in Bangladesh, Applied in Bangladesh, Thailand, and Gujarat. **D & E & F.** Envelope test of simulated SPPs generated with the model trained in Thailand, Applied in Bangladesh, Thailand, and Gujarat. **G & H & I.** Envelope test of simulated SPPs generated with the model trained in Gujarat, Applied in Bangladesh, Thailand, and Gujarat.

corresponding to either observed data or random samples from the LGCP and random point pattern models, and either homogeneous (CS) or heterogeneous (RFS) farm sizes (Table 3).

Clustering or random distributions yielded substantial differences in terms of predicting the probability of large disease outbreaks under two transmission kernels with different spatial ranges. Fig 6 shows that epidemic simulations with LGCP-generated point patterns matched those in the empirical networks more closely than simulations performed in fully random farm distributions.

In the context of short-ranged transmission, it is noteworthy that the empirical and simulated farm distributions exhibited substantial discrepancies in terms of epidemic potential.

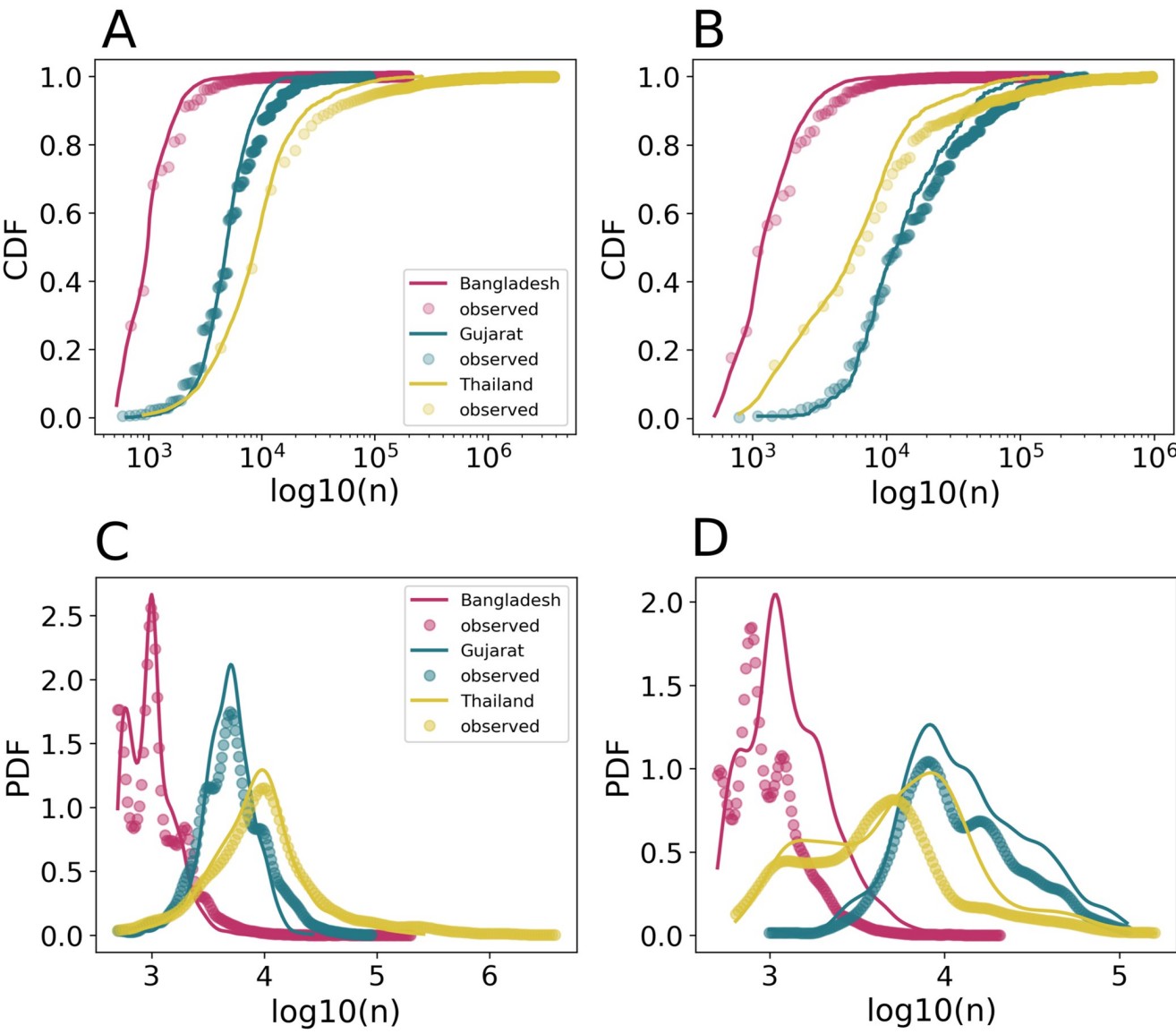

**Fig 5. Distribution of observed and predicted farm size.** Cumulative distribution function ($P(N > n)$) of observed (dots) and predicted (solid line) farm size for broiler farms (A) and layer farms (B). Probability density function of observed (dots) and predicted (solid line) farm size smoothed by a Kernel Density Estimation (KDE) for broiler farms (C) and layer farms (D).

Simulations using random farm distributions significantly underestimated the probability of large outbreaks. In contrast, despite also underestimating the probability of large outbreaks, LGCP models more accurately determined the critical threshold of the transmission parameter ($\beta$), beyond which the risk of an epidemic substantially increases from zero (Fig 6). This suggests that LGCP models are capable of capturing the fundamental dynamics and conditions necessary for disease transmission to occur, even if they somewhat underestimate the overall risk.

Using RF-generated farm sizes or employing a constant (average) value for all farms had minimal impact on the simulations, except in the specific case of the empirical distribution in Thailand (compared using black and grey markers). This discrepancy arose due to the inherent

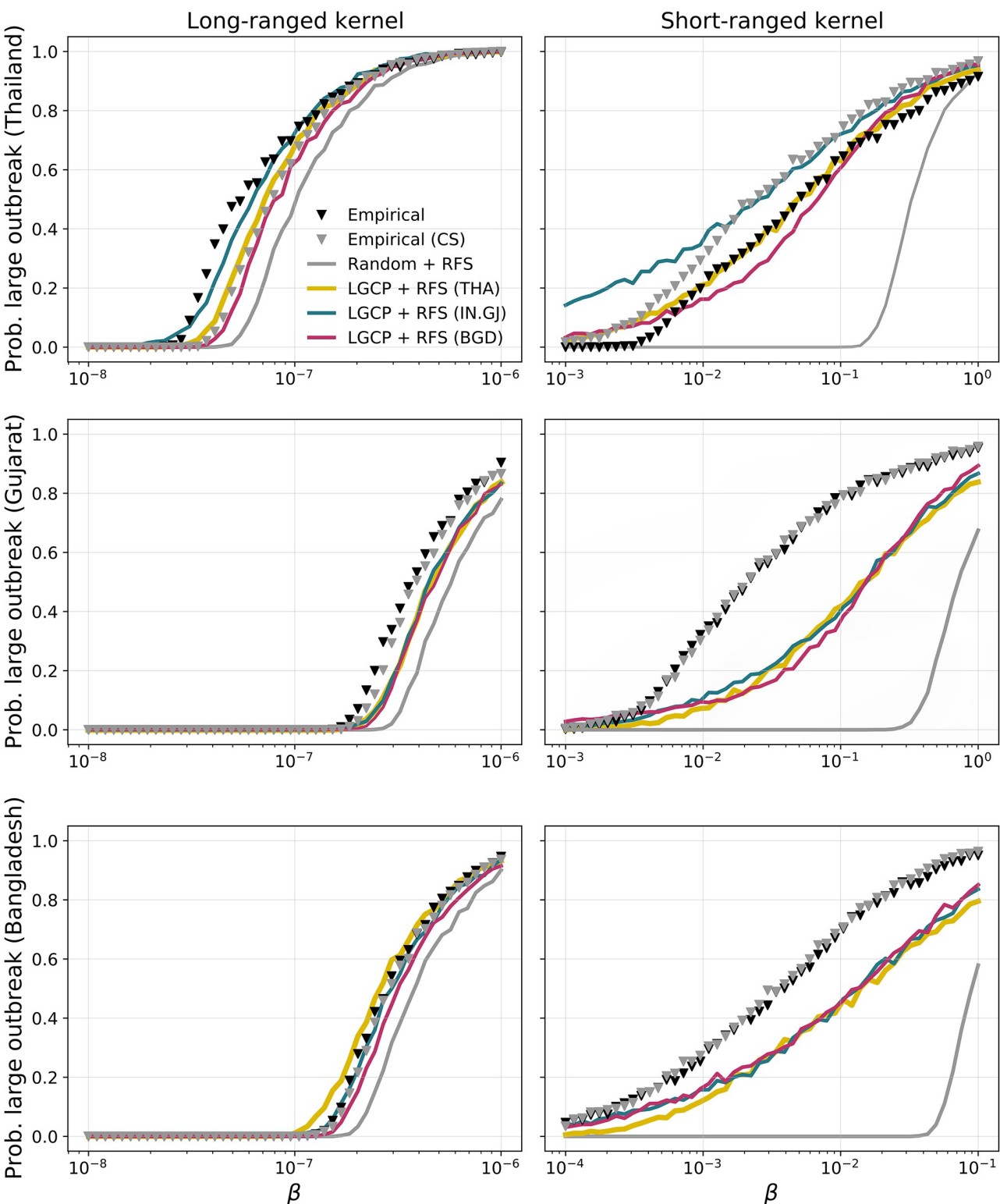

**Fig 6. Probability of large outbreak.** Average probability of large outbreak (*i.e.* the proportion of simulations where the attack rate exceeds 100 farms) as a function of transmissibility for long- and short-range kernels calculated for Thailand (first row), Gujarat (second row) and Bangladesh (last row). The curves shown include LGCP + RFS models trained in Thailand (yellow), Bangladesh (red) and Gujarat (blue). The markers denote simulations using empirical farm locations with original (black) and homogeneous (grey) farm sizes. The grey line correspond to random farm locations with RFS-generated farm sizes. We set $\alpha = 0.643$ for long-range kernel and $\alpha = 3$ for short-range kernel. Other parameters are: $\mu = 0.143d^{-1}$, $Q_S = 1.06$, $Q_I = 0.057$, $d_{min} = 0.1$ km.

heterogeneity of farm sizes across Thailand (Fig 5C), which was not accurately captured by the RF algorithm (S9 Fig). Consequently, when the heterogeneities were mitigated by homogenizing farm sizes, the agreement between LGCP and Empirical+CS (grey) improved significantly.

LGCP models trained in Thailand, Gujarat, and Bangladesh exhibited similar results in Gujarat and Bangladesh, but not in Thailand. Specifically, the model trained in Thailand demonstrated superior performance in the context of the short-ranged kernel, whereas the Gujarat model produced more realistic epidemics when considering the long-ranged kernel. Therefore, it appeared that a single best-performing model cannot be identified based on this analysis.

Spatial risk maps were generated by evaluating the epidemic potential of each farm based on its location (S10 Fig). Boxplots depicting correlation coefficients between risk maps generated with observed and simulated farm distributions are shown in Fig 7. The performance of LGCP models in relation to the long-ranged kernel was robust, as evidenced by their capability to accurately predict the risk maps. Correlation coefficients exhibited some variability with

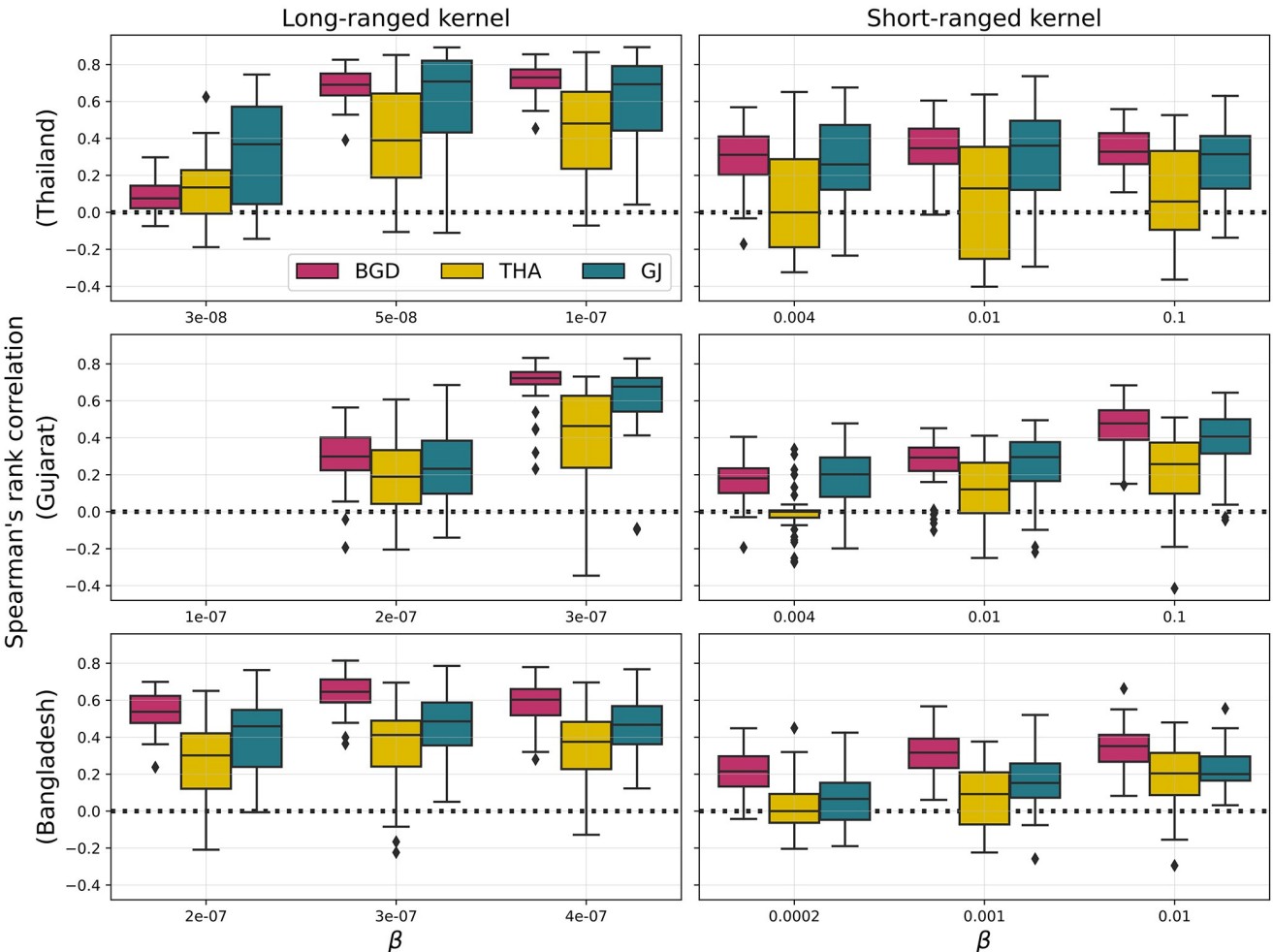

**Fig 7. Spatial risk analysis.** Boxplots show Spearman's rank correlation coefficients between gridded risk distributions for the empirical farm distributions of Thailand (top row), Gujarat (middle row) and Bangladesh (bottom row) and the LGCP+RFS models trained on each area. A single correlation coefficient is calculated for each of 40 realisations from every point pattern model. The first and second columns correspond to long- and short-ranged transmission kernels, respectively. The missing boxplots in the middle-left panel are the due to the fact that no farms became infected when $\beta = 10^{-7}$ and Spearman's correlation coefficient is not defined when all variables in one input set are the same (all equal to 0 in this case).

some simulated configurations displaying risk patterns that were quite different from the observed one. Nevertheless, the median correlation coefficients across all sites are relatively high, suggesting a robust alignment between the predicted risk maps and the actual observed data.

Conversely, correlation coefficients were generally lower for short-ranged transmission, suggesting a decreased predictive accuracy in terms of risk maps. However, it is worth noting that correlation coefficients tended to increase on average with the transmission parameter $\beta$, albeit up to a certain threshold beyond which they reached a plateau.

Remarkably, the model trained in Thailand consistently demonstrated the poorest performance at recovering spatial patterns of epidemic risk across all instances, even when applied to the same country. On the other hand, the model trained in Bangladesh appeared to outperform the other models. Not only did it exhibit higher average correlation coefficients, but it also displayed a narrower range of values, indicating a more consistent and reliable predictive performance compared to the other models, which exhibited greater variability in this regard. These findings highlight the importance of carefully selecting and training LGCP models for specific sites and transmission scenarios, as the choice of training data can significantly impact their predictive capabilities.

## 4 Discussion

Producing accurate spatial maps of livestock farm distributions is paramount for assessing the risk of future epidemics. This study aimed to develop farm distribution models that simulate the locations and sizes of chicken farms, while accounting for production types and spatial clustering of farms in data-scarce countries. The FDM enables the partial reproduction of clustering patterns, the locations of clusters, and farm size in external areas (i.e. when trained in an area and applied to another). In addition, the FDM outputs were used as an input in a disease transmission model to assess whether epidemic patterns are consistent with simulations using observed data.

The LGCP model produces simulated SPPs with farms clustered within specific distances (around <50 to 100 kms for all global envelope tests). The LGCP model, which includes a Gaussian Random Field to induce additional spatial correlation between points [35], allows us to maintain the level of clustering of farms distribution. In comparison, linear regression models and random forest modelling poorly reproduce levels of clustering for intensive livestock production [12]. Indeed, model outputs such as density of animals per pixel do not allow for significant heterogeneity of values, despite employing log transformation of values for model calibration.

Chaiban et al. (2021) [19] were able to reproduce the spatial clustering of farms in the same areas where their model was trained, but failed when considering further regions. In other words, their model failed external validation tests. Indeed, the areas considered in their study were highly heterogeneous in terms of geographical location and level of intensification. Here, we focus instead on areas within South and Southeast Asia with similar degrees of intensification.

We found that the same spatial predictors were able to explain all farm distributions in our study, and lead to acceptable external validation of farm locations compared to the study by Chaiban et al. (2021) [19], where study areas were located in different continents. Similarity in production systems, driven by climatic and economic features, is therefore a crucial factor for choosing appropriate areas for model training. However, the model trained in Thailand provided the worst external validation. This could be caused by the country's geographical characteristics and the specific configuration of the production system, with the country presenting

the highest GDP of our selected study areas. Indeed, economies of scale have shifted the structure of Thai poultry production towards industrialized systems, with fewer producers owning larger holdings [36]. In addition, these structural changes have been supported by the shift from agricultural subcontracting to vertical integration, which involved the centralization of production steps by a few companies and have contributed to the clustering of poultry farms within the peri-urban belt of Bangkok. In contrast, most chickens in Bangladesh are produced in smaller units, which, for most, are not contracted by integrators but rely on credit provided by local production input suppliers to operate [37]. In contrast, in Gujarat, farm sizes are more comparable to Bangladesh but mostly contracted, and not owned, by integrators.

Our study emphasises the importance of accounting for production types when modelling farm distributions, as the level of clustering differs between layer and broiler farms. However, the transition point in the L-function, where the clustering of farms shifts to dispersion, occurs at a similar distance for both broiler and layer farms. This reflects influence of specific country characteristics on these patterns. Indeed, for a given area, the most important predictive variables are the same for both production types.

Epidemic patterns simulated by the disease transmission model align more closely with those obtained using empirical farm distributions under long-range than short-range transmission, particularly in Gujarat. Short-range transmission models are more sensitive to the farms distribution at short distance, therefore their lower efficiency at reproducing similar epidemic patterns might be due to the lower clustering level at short distances in LGCP farm distributions. In Thailand, the discrepancy between observed and simulated farm size distributions likely impacted simulated epidemic patterns. This observation parallels findings from the 1997–1998 Classical Swine Fever epidemic in The Netherlands [38] and epidemic simulations conducted in New Zealand [39], where farms' sizes appeared to affect their susceptibility to infection and infectivity. Nonetheless, the LGCP model outperforms a random distribution and accurately predicts the transmissibility threshold above which a major outbreak becomes probable.

Indeed, spatial clustering increases epidemic risk by lowering said threshold [40, 41]. In highly clustered point distributions, the dynamic of an epidemic strongly depends on the probability of transmission between clusters [42]. Our modelling framework can thus be used to gain insights into the vulnerability of livestock production systems to disease outbreaks under scenarios assuming various levels of clustering and variations in cluster locations. As previously highlighted, when simulating farm distributions in a specific targeted area, the selection of the training area should be based on the similarity of production systems. In cases where there is limited evidence to guide this choice, employing models trained across diverse areas becomes beneficial. This approach generates a spectrum of epidemic patterns capturing uncertainties in the actual underlying distributions of farm locations and sizes.

A limitation of the FDM is that the number of farms cannot be fixed as a parameter, but varies with each simulation. Knowledge about the expected density (i.e. intensity) of farms in the area where the model will be used to simulate farm distributions may be a relevant indicator to select the region used to calibrate the model. Another limitation is that a large number of predictors may lead to an overestimation of the number of farms. Therefore, we tested all combinations of predictors to select the model associated with the largest AUC as an input for the disease transmission model. Assessing the performance of SPP models is non-trivial and still debated in the literature [43]. Our results raise questions about the reliability of the envelope test p-value. Indeed, LGCP models associated with large variations in the level of clustering across simulations will be associated with large p-values as the wide enveloped would include the L-function. We argue that envelope tests should not only be interpreted based on the p-value, but also graphically.

In conclusion, the FDM not only enhances the prediction of data by simulating farm locations and sizes but also significantly improves our understanding of how spatial patterns, particularly farm clustering, influence disease spread. Such understanding is crucial for designing more effective disease control and prevention strategies tailored to local characteristics of production systems. This modelling framework is particularly relevant in resource-limited countries where the intensification of poultry production may often outpace the availability of reliable data. In such contexts, the FDM offers a forward-looking tool, enabling stakeholders to proactively assess epidemic risks associated with various intensification scenarios, and can thus serve as a pivotal tool for informing agricultural planning.

Direct applications of our study include the testing of different levels of clustering and the assessment of their impacts on epidemic spread [42]. By manipulating the degree of clustering in farm distributions and the clusters locations, the vulnerability of livestock systems to disease outbreaks can be explored. Generating such an understanding of the relationship between clustering and epidemic spread is crucial to support policy makers and stakeholders in planning the transition towards safer production systems.

## Supporting information

**S1 Fig. Spatial distribution of layers and broilers farm in Gujarat (2020), Bangladesh and Thailand (2010).** Source of the basemaps: https://gml.noaa.gov/aftp/pub/basu/Borders/GADM/.
(PDF)

**S2 Fig. Illustration of the edge correction for the computation of K-function.** The study area, $W$, is the administrative border of the study region (here, Gujarat is illustrated). The Euclidean distance $d$ between points $x$ and $x'$ is shown, along with a circle of radius $d$ centered at $x$. The portion of the circle, $\ell$, inside the study area is highlighted, demonstrating how the edge correction weight $e_{ij}$ is calculated. This correction accounts for the fraction of the circle's length that lies within the study area, ensuring accurate spatial analysis by adjusting for boundary effects. Source of the basemaps: https://gml.noaa.gov/aftp/pub/basu/Borders/GADM/.
(PDF)

**S3 Fig. Window division and selection for quadrat count test.** Grey tiles were used to calculate the coefficient correlation between observed and simulated points pattern. We did not consider quadrats that occupy less than 80% of the complete theoretical polygon to avoid edge effects. Source of the basemaps: https://gml.noaa.gov/aftp/pub/basu/Borders/GADM/.
(PDF)

**S4 Fig. Short-ranged (blue) and long-ranged kernels (yellow) in a log-log scale.**
(PDF)

**S5 Fig. Scenarios for the epidemiological model in Gujarat.** A. Random farms distribution with constant farm size. B. Farms distribution generated with the LGCP model with constant farm size. C. Random farms distribution with farm sizes predicted with RF model. D. Farms distribution generated with the LGCP model with farm sizes predicted with RF model. Source of the basemaps: https://gml.noaa.gov/aftp/pub/basu/Borders/GADM/.
(PDF)

**S6 Fig. Simulations of point patterns in Gujarat.** A. Observed points patterns of broiler farms in Gujarat and one simulated point pattern with the model trained with Bangladesh broiler farms. B. Mean intensity of points of 8000 simulations from the model trained in

Bangladesh and the observed point pattern is represented with black dots. Source of the basemaps: https://gml.noaa.gov/aftp/pub/basu/Borders/GADM/.
(PDF)

**S7 Fig. Performance of the LGCP.** Internal and external validation p-values of the global rank envelope test for the different models (Bangladesh: BGD, Gujarat: IN.GJ and Thailand: THA), for the two types of farm: broilers (A & C) and layers (B & D) and for radii under 20kms (A & B) and for radii above 20kms (C & D). Labels on the x axis denote the training area. Hatched bars distinguish p-values for internal validation from those for external validation. The color of the bar charts indicate where the model is tested (grey for Bangladesh, blue for Gujarat and yellow for Thailand). The horizontal dashed line indicate the threshold of significance of the p-values for the envelope of 1000 simulations.
(PDF)

**S8 Fig. Farm size modelling.** Predicted farm size in function of observed farm size (A. Broiler and B. Layer). **C & D**. Importance of each covariate for broiler farm RF model (A) and layer farm RF moodel (B).
(PDF)

**S9 Fig. Cumulative empirical size farm distribution (thick line) and farm size distribution from 40 individual realisations of the LGCP+RF model (grey lines) used for the epidemic transmission modellings.**
(PDF)

**S10 Fig. Examples of epidemic risk maps.** Shown for Thailand (top row), Gujarat (middle row) and Bangladesh (bottom row). All simulations are based on a long-ranged transmission kernel and the middle $\beta$ values in Fig 7. The first column shows risk calculated from using the empirical farm distribution. The second and third columns use the average- and best-performing point pattern distributions sampled from the iLGCP+RFS model trained on the same area. Performance is based on Spearman's rank correlation coefficient between gridded risk distributions. White cells contain no farms. Source of the basemaps: https://gml.noaa.gov/aftp/pub/basu/Borders/GADM/.
(PNG)

**S1 File. Algorithm of global envelope test.**
(DOCX)

**S1 Table. Bootstrap analysis of the RF for broiler farms.**
(XLSX)

**S2 Table. Bootstrap analysis of the RF for layer farms.**
(XLSX)

## Author Contributions

**Conceptualization:** Fiona Tomley, Marius Gilbert, Guillaume Fournié.

**Data curation:** Chaitanya Joshi, Madhvi Joshi, Weerapong Thanapongtharm, Madhur Dhingra.

**Formal analysis:** Marie-Cécile Dupas, Francesco Pinotti.

**Funding acquisition:** Fiona Tomley, Guillaume Fournié.

**Investigation:** Marius Gilbert, Guillaume Fournié.

**Methodology:** Marie-Cécile Dupas, Francesco Pinotti.

**Project administration:** Chaitanya Joshi, Madhvi Joshi, Damer Blake, Fiona Tomley, Marius Gilbert, Guillaume Fournié.

**Software:** Marie-Cécile Dupas, Francesco Pinotti.

**Supervision:** Marius Gilbert, Guillaume Fournié.

**Visualization:** Marie-Cécile Dupas, Francesco Pinotti.

**Writing – original draft:** Marie-Cécile Dupas, Francesco Pinotti, Guillaume Fournié.

**Writing – review & editing:** Marie-Cécile Dupas, Francesco Pinotti, Chaitanya Joshi, Madhvi Joshi, Weerapong Thanapongtharm, Madhur Dhingra, Damer Blake, Fiona Tomley, Marius Gilbert, Guillaume Fournié.

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
