## [Decision Letter · Decision Letter 0]

28 May 2024

Dear Dr Dupas,

Thank you very much for submitting your manuscript "Spatial distribution of poultry farms using point pattern modelling: a method to address livestock environmental impacts and disease transmission risks" for consideration at PLOS Computational Biology. As with all papers reviewed by the journal, your manuscript was reviewed by members of the editorial board and by several independent reviewers. The reviewers appreciated the attention to an important topic. Based on the reviews, we are likely to accept this manuscript for publication, providing that you modify the manuscript according to the review recommendations.

Sincerely,

Mark M. Tanaka

Academic Editor

PLOS Computational Biology

Thomas Leitner

Section Editor

PLOS Computational Biology

Reviewer's Responses to Questions

**Comments to the Authors:**

Reviewer #1: In this manuscript the authors develop a method to predict the location and sizes of poultry farms in country with scarce data.

The manuscript is very well written, the objectives are stated clearly and the method they propose is a very interesting approach to tackle the problem of the scarcity of poultry farms data in some countries.

I have only a few comments which I list below:

1) Session 2.3. This is an important part of the method, however the authors don’t provide a clear explanation on how they apply LGCP to simulate the spatial distribution of farms. I understand that they use it to simulate an intensity on a certain area u. But what is the grid size of u? I guess they divide each study region in a grid, with each grid pixel being u. But this is just a guess because I cannot find this information anywhere.

2) Section 2.5.1. In the Random Forest model the authors transform farms size data to reduce the skewness. I was wondering why this is necessary. I am not an expert in random forest models, but I know that random forest is a tree-based model and as such it does not requires scaling of the data.

Reviewer #2: Title: Spatial distribution of poultry farms using point pattern modelling: a method to address livestock environmental impacts and disease transmission risks

This paper speaks to the need for livestock production metrics and the challenge in producing such metrics where data is limited. To address this challenge, the authors develop a (or build upon Chaiban 2021’s) farm distribution model to predict the location and size of poultry farms and test it using farm data from three regions over varying levels of intensification– Bangladesh, Gujarat (a state in India), and Thailand. They use a two-pronged approach to develop the models – first simulating farm locations in point process models and second simulating farm sizes using random forest models. To show the relevance of identifying farm clusters and sizes, they ran simulations of pathogen transmission across different spatial patterns and showed that their distribution model aligned with simulations produced from observed data. This work is important and advances the scientific ability to build higher resolution and more reliable risk analyses that currently use polygon (county, state, etc) level poultry density estimates (such as provided by Van Boeckel 2012, or GLW). The authors’ results show that the non-homogeneity in spatial distribution of farms (and associated farming intensity, or number of animals in production within a farm) is an important component in ensuing risk models. The paper is generally well written, however, I have outlined a number of minor detailed comments that should be addressed below.

Detailed Comments:

1. For future efforts, please provide line numbers within your document to ease the communications in the review process.

2. Page 2, 1st paragraph in Introduction – “In Asia, chicken meat production has quadrupled in the last two decades (FAOSTAT);”. FAOSTAT reference is missing from Lit Cited section.

3. Page 2, 1st paragraph in Introduction ; “this rapid intensification has been characterised

by geographic displacement of farms,”. How has rapid intensification of farming led to geographic displacement of farms? Did original farms close down and move to less rural areas?

4. Page 2, 2nd paragraph: “However, many LMICs do not have the resources needed to keep track of exact farm locations.” These issues also exist for non-LMICs because of privacy issues.

5. References:

Fix formatting of refs, for example:

UK should be capitalized : Keeling, M.J., Woolhouse, M.E., Shaw, D.J., Matthews, L., Chase-Topping,

M., Haydon, D.T., Cornell, S.J., Kappey, J., Wilesmith, J., Grenfell, B.T.,

2001. Dynamics of the 2001 uk foot and mouth epidemic: stochastic dispersal

in a heterogeneous landscape. Science 294, 813–817.

Not full reference: Steinfeld, H., 2006. Livestock’s long shadow: environmental issues and options. Food & Agriculture Org.

6. Page 3, “However, the distribution of animals at farm level and the process of generating clustered point distributions are yet to be embedded in these models.” This has been done for some regions. For example, see Patyk KA, McCool-Eye MJ, South DD, Burdett CL, Maroney SA, Fox A, Kuiper G, Magzamen S. Modelling the domestic poultry population in the United States: A novel approach leveraging remote sensing and synthetic data methods. Geospat Health. 2020 Dec 10;15(2). doi: 10.4081/gh.2020.913. PMID: 33461269.

7. Table 1. The GDP metric is noted for 2019. The text above the table says that data is for 2020. Is it only GDP that is 2019? You could clarify in the title of the table that data is for 2020.

8. Table 2. Title “List of spatial predictors”. Consider having a more complete title.

9. Page 6, The inhomogeneous K-function… “where N is the number of points, |W| denotes total study area, I{dij ≤ r} equals 1 if the euclidian distance dij between points i,j is less than r and is 0 otherwise, and eij is an edge correction weight to avoid sampling biases. Given the location of the first point x and the distance d = ||x − x′||, the second point x′ must lie in the circle b of radius d and centred at x. However, the circle b is generally only partly inside the study area W for large d. Then, the Ripley’s isotropic correction eij uses the fraction of the length of the circle, ℓ, that is ithin the study area and considers that the point pattern is isotropic (statistically invariant under rotation). We calculated the probability of the second point x′ being inside the window W as”… Consider having a supplemental that draws out an example of the spatial application of these equations.

10. Page 7, Global rank envelope test. Please add references for the tests that you used. Similarly, add the R/Python packages used. Indeed, this can be found in your github link for your code, but it would be nice to have it in the main text, as well. It’s listed for some of your methods, but not all.

11. Page 8, Section 2.6.1. What determines a removed farm (R)? Does this mean that a farm had an infection and that once this happens, the model does not allow it to become infected again? Can you please explain this action and the thought process underlying it?

12. Page 9, Section 2.6.2. Can you give more information here on the applicability of the long and short range transmission kernels? Confirming that the alphas were determined by Hill 2017.

13. Figure 1. The description in the title for Panel A is helpful.

14. Page 12, “Crop and human density are important for all models.” According to Fig. 1B, Crop appears as important as the 4 following variables (Access_...) which are the least important except for slope, according to the legend.

15. Fig. 2 title “Envelope test of simulated SPPs generated with the model trained in Bangladesh, in respectively Bangladesh, Thailand and Gujarat.” Please reword this sentence and the one following, as they do not make sense. Also note in the title that the grey shaded areas are the envelope.

16. Page 12, Section 3.2.1. “For broiler farms, the model trained using Bangladesh data offers the best prediction in terms of both internal and external validation. Indeed, the envelopes generated with the model trained on Bangladesh data and applied to Bangladesh and Gujarat include, or are near, the respective observed L-function (Figure 2). Although the Bangladesh model underestimates the clustering level of the Thailand SPP (2B), it reproduces the L-function for low radii of the Gujarat SPPs (Figure 2C) even though it is different from the Bangladesh L-function. Moreover, the Bangladesh-trained model locates the cluster better with high correlation coefficient between observed and simulated SPPs (Figure 3C)”. I follow parts of this section (ie, tighter correlation for models trained on their own data). I do not follow what you are saying for Fig. 3C, nor do I see a C panel in Figure 3. Similar comment through Page 13. Please revise and clarify these sections.

17. Page 16 – similar comment as the preceding comment – I do not see a Panel D for Figure 3.

18. Figure 6. To help the reader more quickly associate the 3 locations, consider outlining the plot borders with the color represented for each location in the other figures (grey, blue, yellow for BGD, GJ, THA, respectively). If you do so, I’d suggest changing the blue color that’s in Fig 4 and others that represent testing data to one that is different than that used for GJ in Fig 3 and elsewhere.

19. Figure 7. Comment for Fig 6 can also be applied here.

20. Page 26, Final paragraph of conclusions: “Direct applications of our study include the testing of different levels of clustering and the assessment of their impacts on epidemic spread (Benincà et al., 2020). By manipulating the degree of clustering in farm distributions and the clusters locations, policy-makers and stakeholders can gain insights into the vulnerability of livestock systems to disease outbreaks. Understanding the relationship between clustering and epidemic spread is crucial for transitioning towards safer production systems.” It is unlikely that decision makers will have the capacity/time to test and manipulate the different degrees of clustering. Are you providing your model as a tool for them to use to do so? This needs to be clarified, if so. The preceding paragraph reads to be the overarching conclusion, but please adjust this paragraph to clarify and strengthen your conclusions.

**Have the authors made all data and (if applicable) computational code underlying the findings in their manuscript fully available?**

Reviewer #1: None

Reviewer #2: Yes

PLOS authors have the option to publish the peer review history of their article (what does this mean?). If published, this will include your full peer review and any attached files.

Reviewer #1: No

Reviewer #2: No

Figure Files:

Data Requirements:

Reproducibility:

References:

---

## [Editor Report · Decision Letter 1]

22 Jul 2024

Dear Dr Dupas,

We are pleased to inform you that your manuscript 'Spatial distribution of poultry farms using point pattern modelling: a method to address livestock environmental impacts and disease transmission risks' has been provisionally accepted for publication in PLOS Computational Biology.

Best regards,

Mark M. Tanaka

Academic Editor

PLOS Computational Biology

Thomas Leitner

Section Editor

PLOS Computational Biology

---

## [Editor Report · Acceptance letter]

12 Aug 2024

PCOMPBIOL-D-24-00410R1 

Spatial distribution of poultry farms using point pattern modelling: a method to address livestock environmental impacts and disease transmission risks

Dear Dr Dupas,

I am pleased to inform you that your manuscript has been formally accepted for publication in PLOS Computational Biology. Your manuscript is now with our production department and you will be notified of the publication date in due course.

With kind regards,

Anita Estes
